# How HIV-1 Nef hijacks the AP-2 clathrin adaptor to downregulate CD4

Xuefeng Ren[1,2,3], Sang Yoon Park[3], Juan S Bonifacino[3]*, James H Hurley[1,2]*

[1]Department of Molecular and Cell Biology, University of California, Berkeley, Berkeley, United States; [2]California Institute for Quantitative Biosciences, University of California, Berkeley, Berkeley, United States; [3]Cell Biology and Metabolism Program, Eunice Kennedy Shriver National Institute of Child Health and Human Development, National Institutes of Health, Bethesda, United States

**Abstract** The Nef protein of HIV-1 downregulates the cell surface co-receptor CD4 by hijacking the clathrin adaptor complex AP-2. The structural basis for the hijacking of AP-2 by Nef is revealed by a 2.9 Å crystal structure of Nef bound to the α and σ2 subunits of AP-2. Nef binds to AP-2 via its central loop (residues 149–179) and its core. The determinants for Nef binding include residues that directly contact AP-2 and others that stabilize the binding-competent conformation of the central loop. Residues involved in both direct and indirect interactions are required for the binding of Nef to AP-2 and for downregulation of CD4. These results lead to a model for the docking of the full AP-2 tetramer to membranes as bound to Nef, such that the cytosolic tail of CD4 is situated to interact with its binding site on Nef.

## Introduction

The human immunodeficiency virus type 1 (HIV-1) is a lentivirus that causes acquired immunodeficiency syndrome (AIDS). HIV-1 has a small genome encoding the main structural proteins Gag, Pol and Env, the regulatory proteins Tat and Rev, and the accessory proteins Nef, Vif, Vpr, and Vpu (*Frankel and Young, 1998*). During viral maturation, Pol is proteolytically cleaved to generate three proteins with enzymatic activity: the viral protease, integrase, and reverse transcriptase. These enzymes are the main targets for chemotherapeutic agents currently in use for the prevention and treatment of AIDS. Combination therapies with these agents have dramatically reduced HIV-1 transmission as well as HIV-1-associated morbidity and mortality. However, concerns about the development of drug resistance in addition to their side effects have fueled a continued search for additional targets. The accessory protein Nef was recognized early on as a potential target for inhibition of the pathogenic effects of HIV-1 (*Coleman et al., 2001*; *Foster and Garcia, 2008*). Although not essential for infection in cell culture, Nef enhances viral replication and disease progression in vivo. The pathogenic effects of Nef are underscored by the observation that patients infected with Nef-deficient strains of HIV-1 often do not develop AIDS for over 10 years even if untreated (these patients are referred to as 'long-term non-progressors' or 'slow progressors') (*Deacon et al., 1995*; *Kirchhoff et al., 1995*; *Gorry et al., 2007*). Inhibition of Nef thus holds the promise to have a similarly beneficial effect. To date, however, this potential has not been realized mainly because Nef has no enzymatic activity and its mechanisms of action are insufficiently understood.

At the cellular level, Nef has been ascribed multiple functions, of which the best characterized and most critical for pathogenesis is the downregulation of CD4 from the surface of infected cells (*Guy et al., 1987*; *Garcia and Miller, 1991*; *Carl et al., 2000*; *Glushakova et al., 2001*). CD4 is a transmembrane protein that acts as a co-receptor in both the host's immune response and the initial binding of HIV-1 to their target cells (*Bowers et al., 1997*). Nef-induced CD4 downregulation interferes with the

*For correspondence:
bonifacinoj@helix.nih.gov (JSB); jimhurley@berkeley.edu (JHH)

Competing interests: The authors declare that no competing interests exist.

**eLife digest** Infection by a pathogen, such as a bacterium or virus, activates both the innate immune response—which is immediate but not specific to the pathogen—and the adaptive immune response, which is stronger and specific to the pathogen. White blood cells called CD4+ T helper cells play an important role in the early stages of the adaptive immune response by helping to activate and regulate other white blood cells that go on to eradicate the pathogen.

HIV-1 is a retrovirus that infects immune cells that have the CD4 receptor on their surface, including CD4+ T helper cells. As the number of worker CD4+ T helper cells falls, the adaptive immune response gradually weakens, and the HIV-1 infected individual becomes increasingly susceptible to infection and disease. An individual is said to develop AIDS when either their CD4+ T helper cell count falls below 200 cells per microliter or they begin to experience specific diseases associated with the HIV-1 infection.

In an effort to prevent and treat AIDS, researchers have worked to understand the HIV-1 genome and have developed medicines that target the enzymatic activity of viral proteins involved in viral replication. When used in combination, these drugs have helped to reduce transmission of HIV-1, and also to reduce deaths from the disease. However, worries about side effects and drug resistance mean that there is a need to develop new drugs.

The HIV-1 genome codes for a number of accessory proteins, including a protein known as Nef that attacks the CD4+ T helper cells, removing the CD4 protein that gives the cells their name. This reduces the ability of the T cells to activate the immune system and allows the virus to spread. Nef acts by forming a complex with a protein called AP-2 in the T cells, and this complex then interacts with the CD4 proteins, causing them to be internalized and then destroyed inside the cells.

Ren et al. have now worked out the structure of the Nef:AP-2 complex at the molecular level and identified the amino acid residues within the Nef protein that interact with the AP-2 protein. This allowed Ren et al. to propose a detailed model of the interaction between the complex and the CD4 protein, and how this leads to the protein being destroyed. This information could be used to develop drugs that work by blocking the amino residues on AP-2 that bind to Nef. Moreover, since these sites are not susceptible to rapid mutations, such drugs are less likely to encounter the problem of drug resistance.

immune system (*Skowronski et al., 1993*), prevents superinfection (*Benson et al., 1993*) and promotes virion release (*Lama et al., 1999*; *Ross et al., 1999*), all of which contribute to enhanced HIV-1 propagation. HIV-1 Nef is a small, polymorphic protein of 200–215 amino acids having a myristoylated N-terminus. X-ray crystallography (*Lee et al., 1996*; *Arold et al., 1997*; *Horenkamp et al., 2011*; *Jia et al., 2012*) and NMR (*Grzesiek et al., 1996a, 1997*) have shown that Nef has a folded core (residues 55–65 and 84–203), with flexible N-terminal (residues 1–54) and C-terminal (residues 204–206) segments, and a central flexible loop (residues 149–179) (residue numbers correspond to the NL4-3 strain of HIV-1). CD4 downregulation depends on both Nef myristoylation (*Aiken et al., 1994*) and specific residues in the loop, including Leu164 and Leu165 (*Bresnahan et al., 1998*; *Craig et al., 1998*; *Greenberg et al., 1998*; *Janvier et al., 2003*), which are in a sequence context fitting the [DE]XXXL[LI] motif for dileucine-based sorting signals (*Bonifacino and Traub, 2003*), and the diacidic motif, Asp174-Asp175 (*Aiken et al., 1996*; *Lindwasser et al., 2008*). Myristoylation allows recruitment of Nef from the cytosol to the inner leaflet of the plasma membrane (*Yu and Felsted, 1992*) while the loop engages the clathrin-associated adaptor protein 2 (AP-2) complex (*Jin et al., 2005*; *Chaudhuri et al., 2007*; *Doray et al., 2007*; *Lindwasser et al., 2008*; *Mattera et al., 2011*; *Jin et al., 2013*). The Nef:AP-2 complex interacts with the cytosolic tail of CD4, leading to the cooperative assembly of a tripartite Nef:AP-2:CD4 complex (*Chaudhuri et al., 2009*). This complex nucleates the formation of clathrin-coated pits (*Foti et al., 1997*; *Greenberg et al., 1997*; *Burtey et al., 2007*) that mediate rapid internalization of CD4, followed by its delivery to lysosomes via the multivesicular body pathway (*Aiken et al., 1994*; *Rhee and Marsh, 1994*; *daSilva et al., 2009*).

Biochemical analyses have demonstrated that binding of Nef to AP-2 is direct and dependent on the dileucine and diacidic motifs, and other residues, in the Nef loop (*Chaudhuri et al., 2007*; *Doray et al., 2007*; *Lindwasser et al., 2008*; *Chaudhuri et al., 2009*; *Mattera et al., 2011*). AP-2 is a

heterotetramer composed of α, β2, μ2 and σ2 subunits. The N-terminal 'trunk' domains of α and β2 together with the whole μ2 and σ2 subunits constitute the core of the complex, whereas the C-terminal 'hinge' and 'ear' domains of α and β2 form long projections that extend from the core (*Owen et al., 2004*). The AP-2 core undergoes a large conformation change from a 'locked' to an 'open' conformation that allows it to bind sorting signals and to be recruited to membranes via interaction with the phosphatidylinositol lipid PI(4,5)P$_2$ (*Jackson et al., 2010*). Nef has been shown to bind to the α–σ2 'hemicomplex' (*Chaudhuri et al., 2007*; *Doray et al., 2007*). The σ2 subunit (with a small contribution from the α subunit), harbors a binding site for [DE]XXXL[LI]-type signals from host cell proteins (*Kelly et al., 2008*; *Jackson et al., 2010*; *Mattera et al., 2011*). Mutational analyses have shown that this site is also required for Nef binding, most likely through recognition of the Nef dileucine motif (*Mattera et al., 2011*). The α subunit has an additional site, comprising Lys298 and Arg341, which is also required for Nef binding and CD4 downregulation (*Chaudhuri et al., 2009*). Although it is tempting to hypothesize that these basic residues interact with the Nef diacidic motif, there is currently no direct evidence for such an interaction. Importantly, this second site on α is not known to participate in any host cell function, making it a possible target for selective interference.

Despite progress in the identification of determinants of the Nef-AP-2 interaction, the conformation of the Nef loop when bound to AP-2 and the molecular details of the interaction are not known. To elucidate the structural basis for this interaction, we have solved the crystal structure at 2.9 Å resolution of Nef (residues 54–203) in complex with the α (residues 1–396) and σ2 (full-length) subunits of AP-2. The structure reveals that the entire central loop is well ordered, and that most of it contacts the α–σ2 hemicomplex. The Nef core is directly involved in contacts as well as serves as a scaffold to position the central loop. The structure leads to a model for the docking of HIV-1 Nef onto the plasma membrane in conjunction with AP-2, and suggests how the AP-2:Nef complex binds to the CD4 cytosolic tail in the membrane setting.

## Results

### High-affinity binding of HIV-1 Nef to the α–σ2 hemicomplex

In the absence of PI(4,5)P$_2$-containing membranes, the AP-2 core is in a locked conformation that has low affinity for both physiological cargoes and Nef. Previously, we assayed a version of the AP-2 core in which the μ2 C-terminal domain was deleted so as to destabilize the locked conformation. This construct bound to HIV-1 Nef with $K_d$ = 6 μM as judged by surface plasmon resonance (*Chaudhuri et al., 2007*). However, the conformational lability introduced into this construct made it unsuitable for crystallization. We built on the finding that Nef interacts with the α–σ2 hemicomplex as judged by yeast three hybrid (Y3H) (*Chaudhuri et al., 2007*) and pulldown assays (*Doray et al., 2007*). Hemicomplex constructs including the full α trunk domain are poorly stable, because a large amount of hydrophobic surface area is exposed on the C-terminal part of the trunk domain when the hemicomplex is excised from the intact AP-2 core. A truncated version of the homologous γ-ζ hemicomplex of COPI including the first 17 helices of the γ trunk domain was found to be suitable for crystallography (*Yu et al., 2012*). We designed a construct comprising the first 19 helices (residues 1–396) of the α trunk domain and co-expressed it with full-length σ2 (*Figure 1A,B*). This portion of the α trunk includes all of the Nef-interacting residues of α that have been documented to date. This construct bound to the HIV-1 NL4-3 Nef (54–203) (hereafter, 'Nef') with $K_d$ = 1.8 μM and 1:1 stoichiometry, as determined by isothermal titration calorimetry (ITC) (*Figure 1C*). The comparatively high affinity of the interaction and the congruence with previous results with the tetrameric construct led us to conclude that this hemicomplex included all the major determinants of the AP-2:Nef interaction.

### Crystal structure of the α–σ2:Nef complex

Nef was co-crystallized with the α1–396 form of the AP-2 α–σ2 hemicomplex (hereafter, 'α–σ2'). The structure was determined by molecular replacement at 2.9 Å resolution (*Figure 2A,B*; *Table 1*). The asymmetric unit contains four Nef:α–σ2 complexes, all in similar conformations, with small variations in the quality of the electron density. The description will focus on the B, C, and D chains, for which the Nef:α–σ2 interface is most clearly visualized. Nef buries 1170 Å$^2$ in this interface, of which two-thirds is buried against σ2 and the remainder against α. The α–σ2 unit is essentially rigid, in a conformation identical to that seen in other structures of the AP-2 complex (*Collins et al., 2002*; *Kelly et al., 2008*; *Jackson et al., 2010*). The Nef core (excluding the central loop 149–179) contains the five α-helices

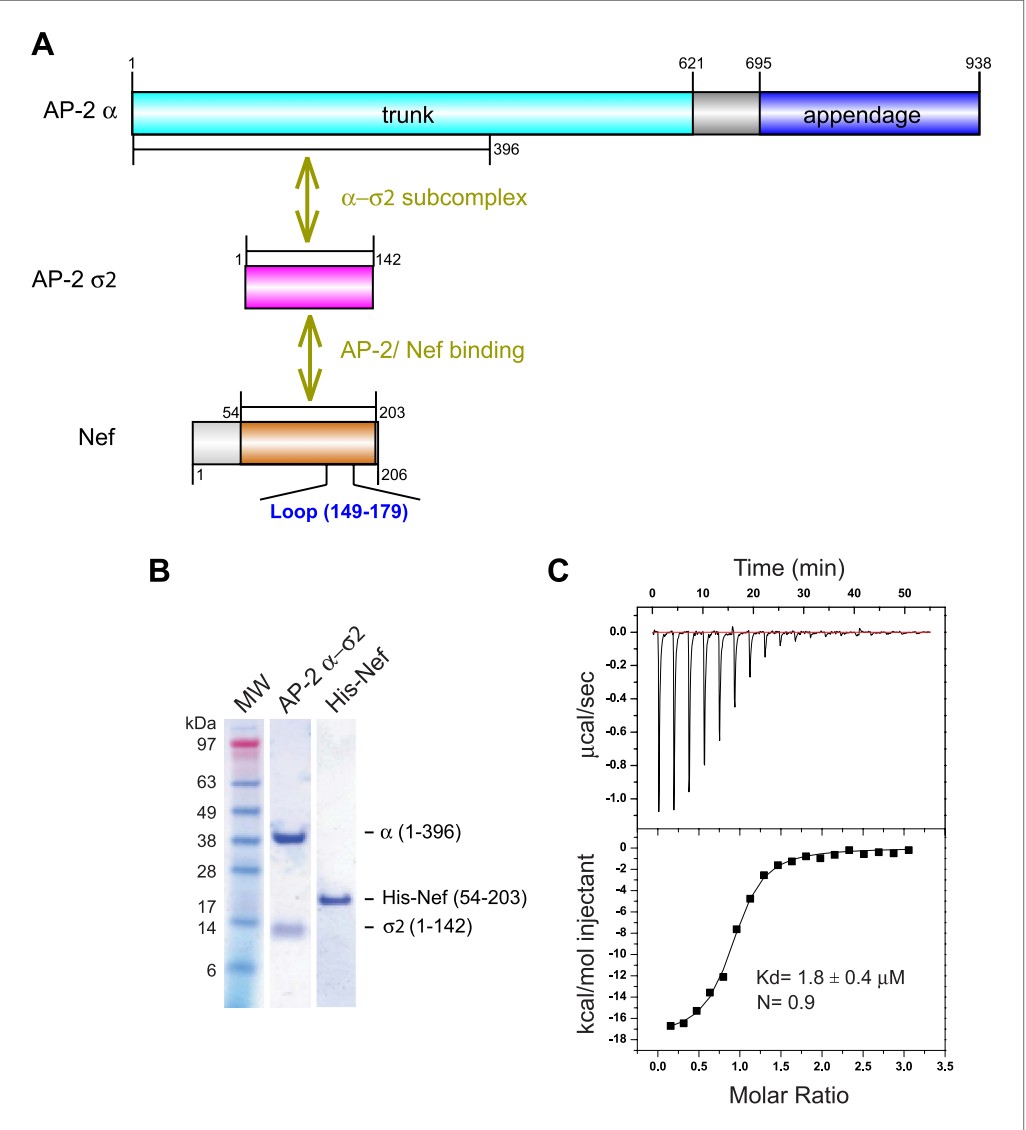

**Figure 1**. Nef binds with low micromolar affinity to the AP-2 α–σ2 hemicomplex. (**A**) Schematic representation of AP-2 α–σ2 and Nef protein constructs. AP-2 α (1–396) (cyan) and full-length σ2 (magenta) were generated as a stable subcomplex and the interaction with the indicated Nef construct (54–203) (orange) was analyzed. (**B**) SDS gel of recombinant AP-2 α–σ2 and Nef proteins. (**C**) Isothermal titration calorimetry of the titration of His-tagged Nef (54–203) to the AP-2 α–σ2 hemicomplex. The upper panel shows the differential heat released when Nef (0.6 mM) was injected into AP-2 α–σ2 solution (40 μM) in 2.1 μl aliquots.

(H2, H3, H6, H7, H8) and five-stranded β-sheet visualized in other Nef crystal structures (*Lee et al., 1996*; *Figure 2C*). The core also manifests a poorly ordered N-terminal helix (H1) spanning residues 55–65, which was not visualized in all of the chains. The identity of this helix was provisionally assigned on the basis of the only other crystal structure in which this region was visualized (*Breuer et al., 2011*). This helix was first identified by solution NMR and contains the primary binding site for CD4 (*Grzesiek et al., 1996a*). In contrast to most other crystal structures, the central loop from residues 149–179 was visualized in its entirety (*Figure 2A*). The central loop contains two additional helices, one from residues 150–157 (H4), and the other a single turn from 167–170 (H5). The central loop interacts extensively both with the α and σ2 subunits (*Figure 2D*), with the greatest contact surface involving σ2. The core interacts primarily via a network of interactions between helix H3 and the α subunit.

The landmarks within the central loop are helix H4 (150–157), the acidic-dileucine motif (160–165), helix H5 (167–170), and finally a series of turns centered on Met173 (171–179) (*Figure 2D*).

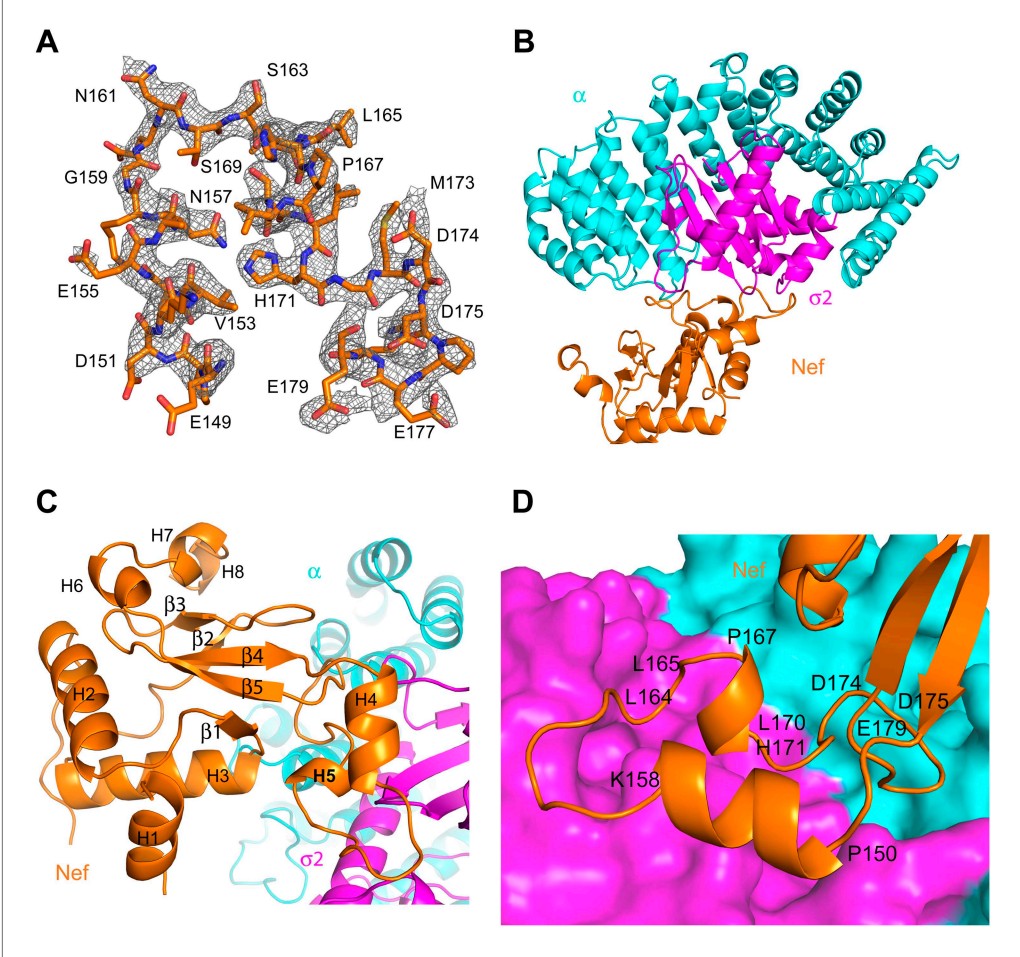

**Figure 2**. Crystal structure of the AP-2 α–σ2:Nef complex. (**A**) $F_0$-$F_c$ omit map of Nef loop (149–179) with the final model superimposed. The map is contoured at 2.0 σ. (**B**) Overall ribbon representation of AP-2 α (cyan) and AP-2 σ2 subunits (magenta) in complex with Nef (orange). (**C**) Detailed ribbon model of Nef (orange) with the secondary structures indicated. (**D**) Ribbon model of the Nef central loop (149–179), which includes helix H4 (150–157), the acidic-dileucine motif ($_{160}$ExxxLL$_{165}$), helix H5 (167–170), and the C-terminal turn-rich segment (171–179).

The intramolecular interactions of H4 are with other sections of the central loop, explaining why this helix has not been observed in other structures of Nef. The Leu164-Leu165 pair of the dileucine motif anchors the loop in a pocket on σ2 just as seen for a dileucine peptide bound to the unlatched AP-2 tetramer (**Kelly et al., 2008**; **Figure 3A**). The dileucine peptide in the unlatched structure (**Kelly et al., 2008**) used for comparison is derived from CD4, but it is important to emphasize that this binding mode is dependent on phosphorylation and is unrelated to Nef-dependent downregulation. The pocket walls are formed by hydrophobic residues of σ2. Nef Glu160 of the motif binds to basic residues on both σ2 and α (**Figure 3B**). Residues of H4, notably Glu154, make electrostatic interactions with a second basic patch on σ2 (**Figure 3C**). H4 and the dileucine motif have little or no interaction with the Nef core, and their conformation seems to be specified by their interactions with α–σ2.

In contrast, helix H5 and the C-terminal turn segment are sandwiched between α–σ2 and the Nef core. Their structure clearly depends on the interactions with α–σ2, since they are not otherwise visualized in this conformation. However, the Nef core also seems to have an important role in organizing this segment. H5 packs against the β-sheet of the core, and serves primarily to orient the hydrophobic loop with respect to the dileucine motif. The tight turns of the C-terminal part of the loop serve to project a number of charged and hydrophobic side chains into complementary interactions with both α and σ2 (**Figure 3D**). This unusual sequence of turns is anchored at its ends by H5 and by the strand

**Table 1.** Statistics of crystallographic data collection and refinement

| Construct | AP-2: σ2 (1–143), α (1–396); Nef (54–203) |
|---|---|
| Data collection | |
| X-ray source | APS 22-ID |
| Wavelength (Å) | 1.0000 |
| Space group | $P2_12_12_1$ |
| Cell dimensions | a = 109.56 Å, b = 168.03 Å, c = 200.20 Å, α = β = γ = 90° |
| Resolution (Å) (last shell) | 50.00–2.90 (3.00–2.90) |
| Unique reflections | 80,188 |
| $R_{sym}$* (%) | 18.4 (53.9) |
| I/σ | 7.4 (1.9) |
| Completeness | 95.8 (80.2) |
| Redundancy | 5.2 (3.2) |
| Refinement | |
| $R_{work}$/$R_{free}$ (%) | 21.9/26.7 |
| Average B values (Å²) | 39.5 |
| Number of protein atoms | 21,588 |
| R.m.s. bond length deviation (Å) | 0.015 |
| R.m.s. bond angle deviation (°) | 1.16 |
| Ramachandran Plot (%) | |
| Favored | 98.4 |
| allowed | 1.2 |
| outlier | 0.4 |

*$R_{sym} = \Sigma_h \Sigma_i |I_i(h) - <I>| / \Sigma_h \Sigma_i I_i(h)$, where I is the observed intensity and <I> is the average intensity of multiple observations of symmetry-related reflections.

β5 of the core. The whole turn-rich section of the loop from 171–179 is anchored internally by a hydrogen bond between Nef Asp174 and the main-chain amide of Gln104, and by a partially buried salt bridge between Nef Asp175 of the loop and Nef Arg134 of the core β sheet (*Figure 3E*).

The structure also reveals that residues of Nef helix H3 of the core directly contact AP-2. In particular, Gln104, Arg105, and Asp108 bind to a basic patch on α (*Figure 3F*). This polar interface adjoins the mixed polar and hydrophobic interface created by the C-terminal turn segment of the central loop. There is one other minor interaction with the core region, involving Nef Pro129 of the β2-β3 loop. The Pro side-chain forms van der Waals interactions with atoms of the α Arg341 side-chain. While the central loop clearly dominates the interactions overall, the Nef core interactions are also significant, and represents one of the completely unexpected findings from the structural analysis.

## Nef residues in AP-2 binding and CD4 downregulation

The Nef-AP-2 interaction is so central to CD4 downregulation that it has inspired exhaustive mutational analyses (*Aiken et al., 1996*; *Hua et al., 1997*; *Craig et al., 1998*; *Janvier et al., 2003*; *Lindwasser et al., 2008*; *Chaudhuri et al., 2009*; *Mattera et al., 2011*; *Jin et al., 2013*). These results can now be mapped onto the structure (*Figure 4*). We performed additional mutagenesis to test for the functional importance of residues that were newly identified by the structure determination (*Figures 5 and 6*). Mutations in the regions of AP-2 that were already known to bind dileucine signals had the expected loss of interaction. These include σ2 Y62A and A63D (*Figure 5C*). Mutations in regions unique to Nef binding, notably σ2 R60E (*Figure 5C*) and α E342K (*Figure 5D*), also eliminated binding. Other nearby residues with more peripheral interactions, including σ2 N48A, H85A, and C99A and α V300A, Q301A, and N344A, had lesser mutational phenotypes, if any (*Figure 5C,D*). The results of these analyses are represented in *Figure 4* together with previously published data. The collective body of work and its structural mapping are summarized in *Table 2*. The large majority of the mutational hits map to

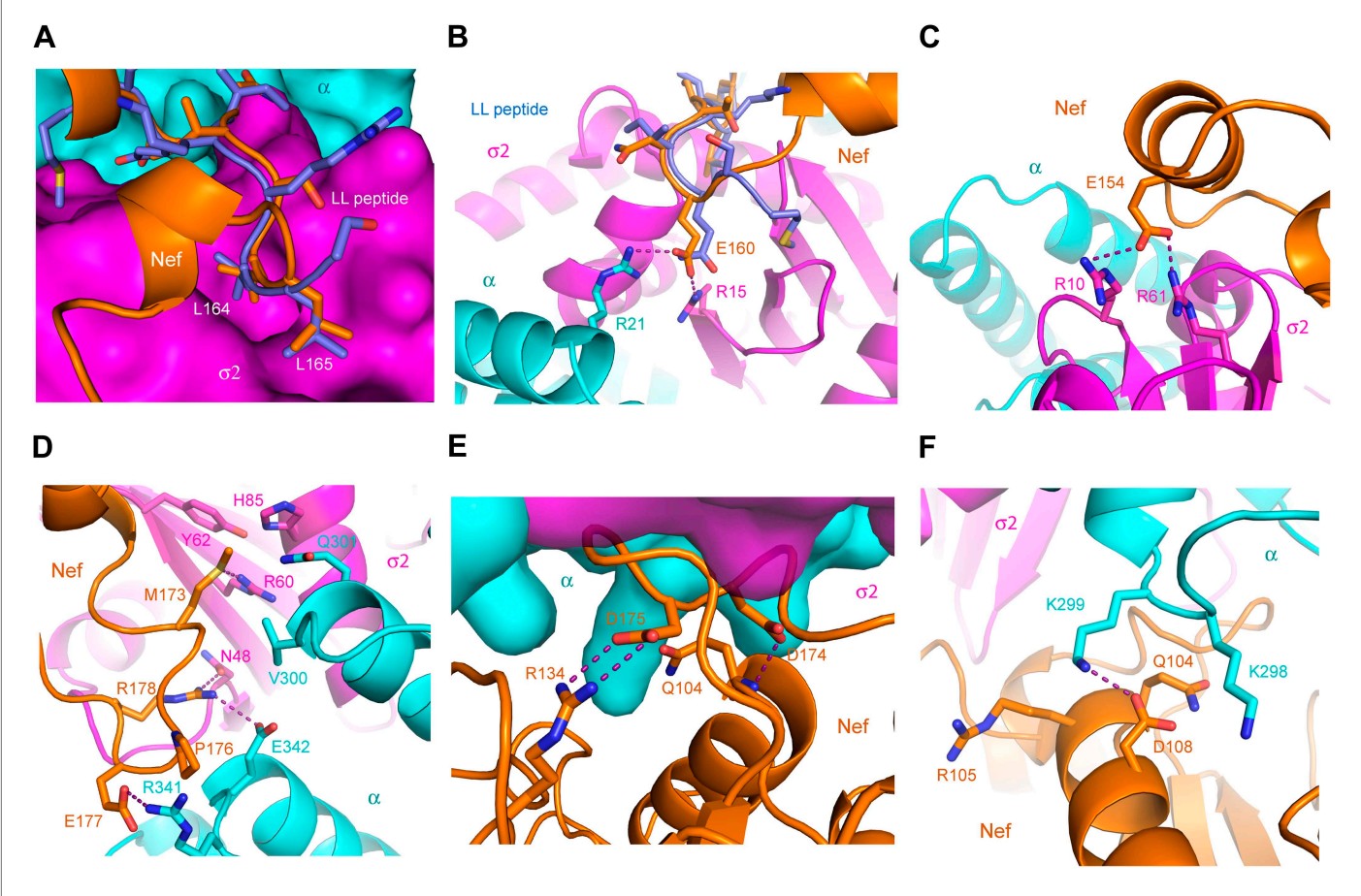

**Figure 3**. The AP-2 α–σ2:Nef interface. (**A**) Stick representation of the Nef dileucine motif (Leu164 and Leu165, orange) interacting with AP-2 σ2 (magenta), compared with a bound dileucine peptide (blue, PDB id: 2JKR) (**Kelly et al., 2008**). (**B**) Nef Glu160 of the acidic-dileucine motif forms hydrogen bonds with AP-2 α R21 and σ2 R15. The hydrogen bond is listed as a purple dashed line. (**C**) Nef Glu154 in helix H4 (orange) forms hydrogen bonds with AP-2 σ2 R10 and R61 (magenta). (**D**) The C-terminal part of Nef loop (171–178) interacts with both AP-2 α and σ2 (magenta). (**E**) The key Nef diacidic motif Asp174 and Asp175 forms intramolecular hydrogen bonds that stabilize the loop conformation. Hydrogen bonds occur between the side chain of Asp174 and the main-chain amide NH of Gln104, and between the side-chains of Nef Asp175 and Arg134. (**F**) A salt bridge between Asp108 of Nef helix H3 bridges the Nef core to a basic patch on α.

residues that directly participate in Nef-AP-2 contacts. The consistency validates both the previous mutational approach and the structural findings.

Previous analyses had highlighted the importance of a basic patch on α comprising both Lys298-Lys299 and Arg341. The present structure revealed that Nef Glu177 is the primary interaction partner for α Arg341. The Nef E177K mutant manifested a reduction in both α–σ2 binding (**Figure 5B**) and CD4 downregulation (**Figure 6A,B**), consistent with its structural role. The deepened insight obtained from the crystal structure shows that this patch is better conceived of as a polar, rather than basic, patch. For example, the side chain of α Glu342 has a close approach to Nef Arg178.

The finding that Nef Asp174 and Asp175 do not contact AP-2 directly was a surprise. These two residues are required for AP-2 binding (**Figure 5B**) and CD4 downregulation (**Lindwasser et al., 2008**), and had been expected to interact with a basic patch on α. They are, in fact, in the general vicinity of a basic patch, close to α Lys298 and Lys299. Nevertheless, Asp174, while close to the interface, is not in direct contact with AP-2. Its key role appears to be to anchor the turn section to the N-terminus of helix H3. The partial positive charge at the helix N-terminus can form an interaction with Asp side-chains that is almost as energetically favorable as a salt bridge. In this case, a short hydrogen bond is formed with the main-chain amide group of Nef Gln104. Asp175 has a similar role in conformational stabilization. Asp175 is partially buried in a contact with the Nef core, and forms a salt bridge with Arg134.

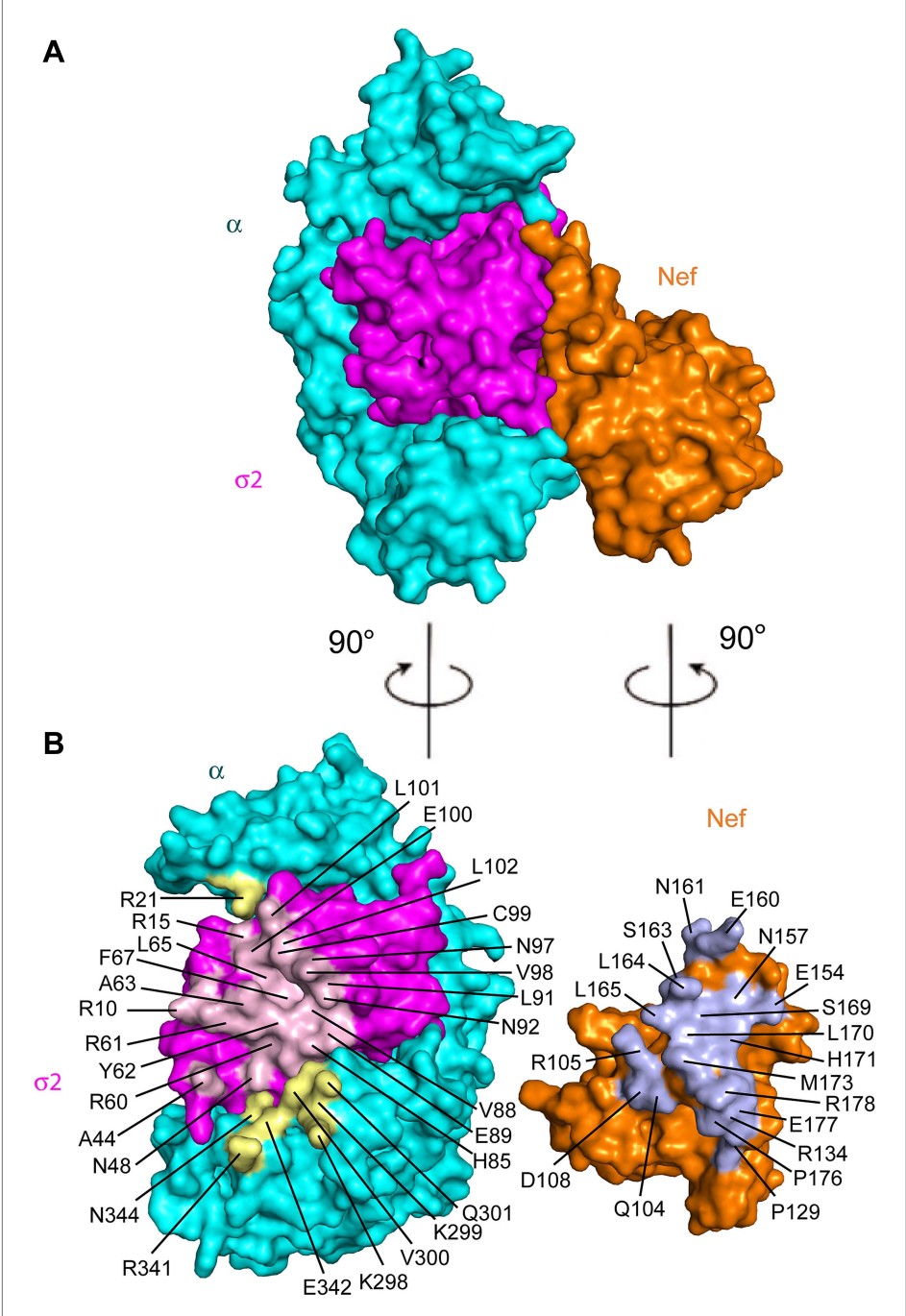

**Figure 4**. Structural mapping of mutations that interfere with binding and CD4 downregulation. (**A**) The surface representation shows the contact between AP-2 α–σ2 and Nef. (**B**) AP-2 α–σ2 or Nef interfaces are rotated by 90° to expose the interaction surfaces directly to view. Interacting residues in AP-2 α are colored in yellow, residues in AP-2 σ2 are colored in pink, and residues in HIV-1 Nef are highlighted in light blue.

We hypothesized that the internal Nef Arg134-Asp175 internal salt bridge is important for stabilizing the C-terminal turn portion of the central loop in its AP-2 binding conformation. Indeed, mutation of Arg134 to Glu abrogated interaction of Nef with α–σ2 (*Figure 5B*). Moreover, co-expression in HeLa cells of CD4 with Nef R134E followed by FACS analysis showed that this Nef mutant almost completely lost its ability to downregulate CD4 (*Figure 6A,B*). These results phenocopy the D175A change

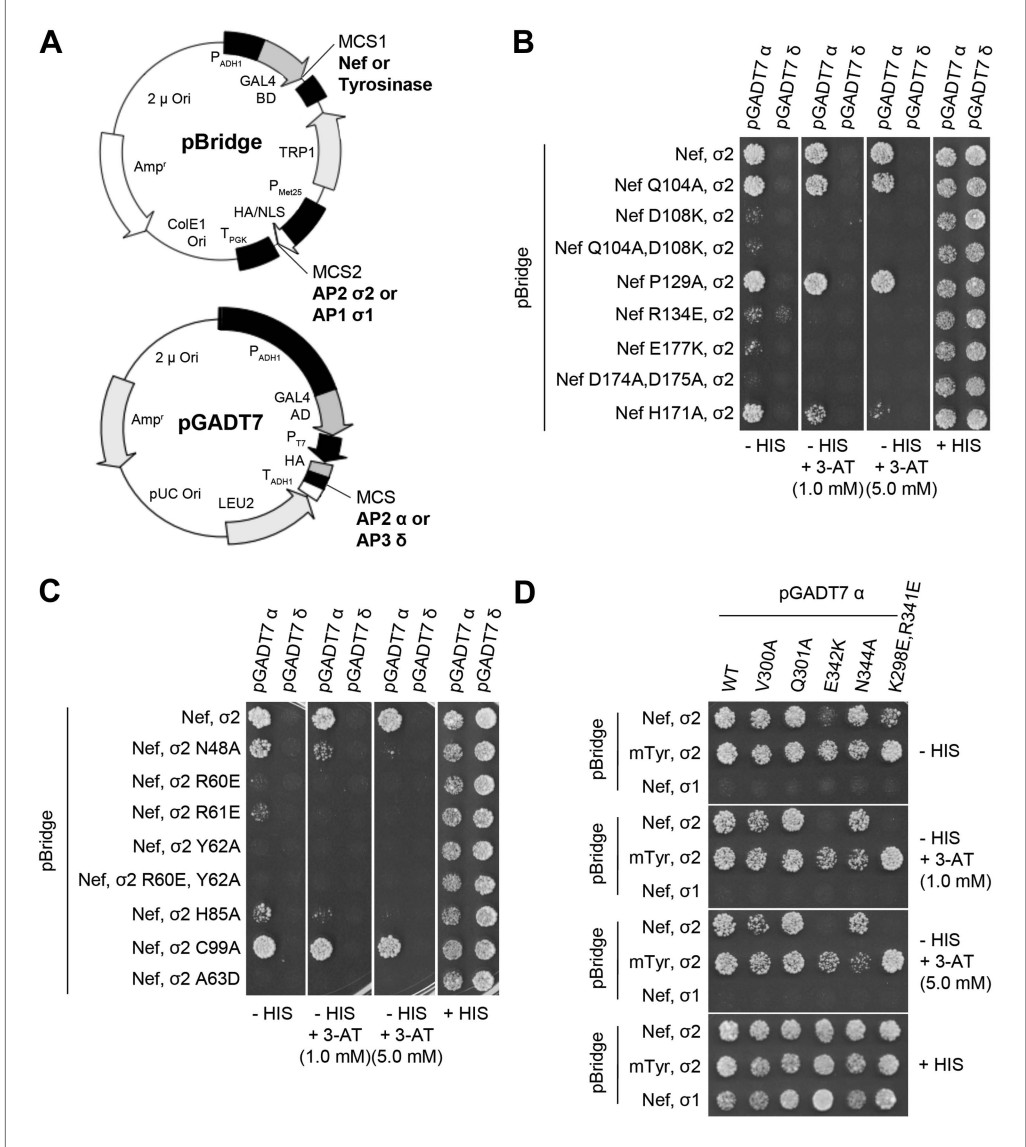

**Figure 5**. Structural interface mutants of Nef and AP-2 complexes prevent binding. Y3H analysis of HIV-1 Nef and AP-2 α–σ2 hemicomplexes with mutations of residues revealed in the crystal structure. (**A**) Diagram of the plasmids used in Y3H analysis. NL4-3 Nef or mouse Tyrosinase cytosolic tail was cloned into MCS1 of pBridge and expressed as a GAL4BD fusion protein. AP-2 σ2 or AP-1 σ1 was cloned into MCS2 of pBridge and expressed without Met. AP-2 α or AP-3 δ was cloned into MCS of pGADT7 and expressed as a GAL4AD fusion protein. (**B–D**). The indicated combinations of double transformants were plated in media lacking Leu, Trp, Met and His (−HIS),−HIS with 3-AT (1 mM or 5 mM) or Leu, Trp and Met (+HIS). mTyr, mouse Tyrosinase cytosolic domain.

(*Lindwasser et al., 2008*) and are consistent with a critical role for the salt bridge in stabilizing the conformation of the Nef loop required for AP-2 binding and CD4 downregulation.

While most mutational studies of the Nef-AP-2 interaction focused on the Nef central loop, one face of helix H3, including residues Gln104 and Asp108, contacts AP-2. The Nef mutant Q104A behaves like wild type in both binding to α–σ2 (*Figure 5B*) and CD4 downregulation (*Figure 6A,B*). This mutation does not alter the main-chain, and so will not affect the ability of the main-chain of residue 104 to help anchor the central loop via Asp174. The charge-reversal mutant D108K, however, eliminates α–σ2 binding as judged by Y3H (*Figure 5B*) and CD4 downregulation (*Figure 6A,B*). This finding is consistent with the salt bridge seen between α Lys299 and Nef Asp108 in the structure. Another core residue, Pro129, has limited van der Waals interactions with α, thus it was not surprising

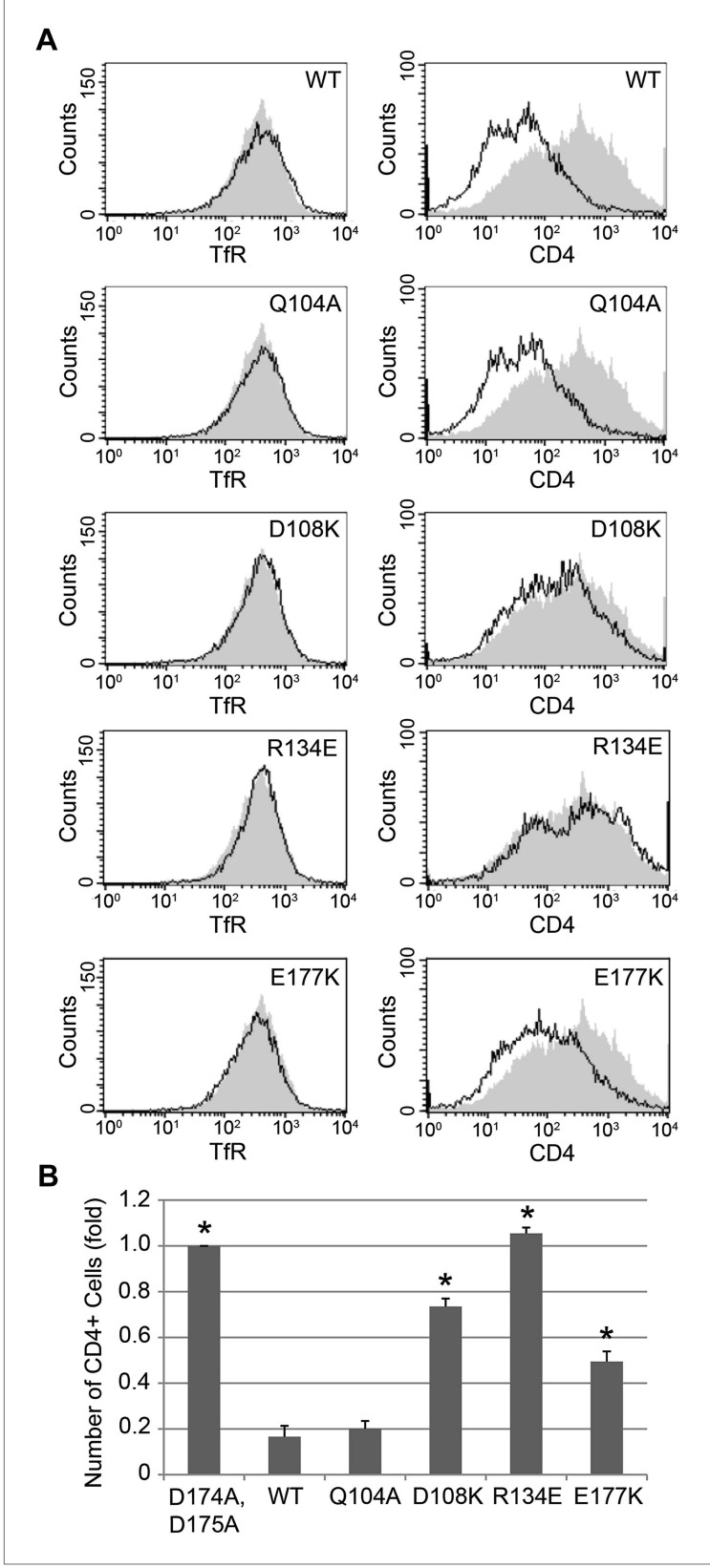

**Figure 6**. Nef interface mutants do not downregulate CD4. Nef Asp108, Arg134, and Glu177 are required for the Nef-induced CD4 downregulation. (**A**) HeLa cells were cotransfected with pCMV-CD4 and pIRES-eGFP-Nef wild-type or mutant plasmids for 24 hr. The cells were then stained with APC-conjugated anti-CD4 antibody and

*Figure 6. Continued on next page*

*Figure 6. Continued*

PE-conjugated anti-Transferrin receptor (TfR) antibody. GFP was used as an indicator for transfected cells. The D174A, D175A mutant Nef was used as a negative control (Shaded curves in all plots). Data shown are representative of three independent experiments. (**B**) The graph shows the relative number of CD4 positive cells from *Figure 6A* (mean ± SD; N = 3; asterisks: p<0.001 compared with wild-type Nef).

that its mutation has no effect on the interaction (*Figure 5B*). These results corroborate the importance of core helix H3 in the AP-2 interaction and CD4 downregulation.

## The membrane-bound AP-2:Nef complex

AP complexes function as membrane-bound, tetrameric assemblies. A consensus view of the structure and membrane-docking mode of AP complexes has emerged from the structures of active conformations of the tetrameric AP-1 and AP-2 cores (*Jackson et al., 2010*; *Canagarajah et al., 2013*; *Ren et al., 2013*). A model for the membrane-bound AP-2:Nef complex in the consensus docking geometry was generated by superposition of the α–σ2 hemicomplex on the full open tetramer (PDB 2XA7) (*Jackson et al., 2010*). A steric clash was observed with helix 1 of the β2 subunit. β2 helix 1 was previously shown to be conformationally labile in the unlatching (partial activation) of AP-2 (*Kelly et al., 2008*), and it seems reasonable to expect that it pivots in the full AP-2:Nef complex to prevent a collision.

The consensus docking of the AP-2:Nef complex places Nef proximal to the membrane such that the N-terminal helix H1 is aligned parallel and in contact with the membrane (*Figure 7A,B*). The C-terminus of helix H2 and parts of the central loop also align such that they would contact the membrane surface. The N-terminus of H1 is exposed to solvent such that there would be nothing to impede contact of the myristoylated N-terminus with the membrane. Indeed, the exposed face of H1 conjoins with H2 and parts of the central loop of Nef and the membrane-proximal parts of AP-2 to form a bowl that is 30 × 30 Å across, with a clearance of ~10 Å from the membrane (*Figure 7C*). Strikingly, the exposed face of H1 contains residues Trp57 and Leu58 that have been implicated in direct binding to CD4 (*Grzesiek et al., 1996a*; *Figure 7D*). Other residues implicated in CD4 binding, including Leu97, Arg106, and Leu110, also project into the bowl (*Grzesiek et al., 1996a*). The edge of the bowl includes residues from α, σ2, and β2 (*Figure 7C,D*). This suggests that multivalent interactions between Nef, CD4, and AP-2 likely drive cooperativity in the formation of the ternary complex.

## Discussion

The structure of the AP-2:Nef complex provides a framework to unify nearly two decades' worth of research of the molecular basis for CD4 downregulation by Nef. The large majority of Nef residues that have been implicated in CD4 downregulation reside in the central loop. The structure shows that all of these residues are well ordered in the complex, in contrast to all previous structures of Nef–effector complexes. Nearly all of these Nef residues directly contact AP-2. One notable exception is Asp175, which had been anticipated to bind to a basic patch on the surface of the α subunit. The structure revealed an unexpected role for Asp175 in stabilizing the conformation of the central loop. The structure also shows that the Nef core has both a direct role in forming polar interactions with AP-2 and an indirect role in scaffolding the conformation of the central loop. The role of Asp174 and Asp175 amounts to the formation of an effector-specific polar core within Nef. This provides insight into the underlying reason for the unusual architecture of Nef, as a single domain protein with disproportionately large internal loops. This architecture gives Nef an exceptional degree of plasticity, allowing multiple functions to be encoded within a relatively small structure.

The structure is beautifully consistent with the emerging consensus picture that all activated AP complexes seem to bind to membranes in the same conformation and the same geometry (*Jackson et al., 2010*; *Canagarajah et al., 2013*; *Ren et al., 2013*). The membrane-docking geometry suggested by the unlocked states of AP-1 and AP-2 places Nef such that it is touching the membrane, with its N-terminal region membrane proximal. This is consistent with its essential N-terminal myristoylation. One question still to be resolved is the disposition of the first helix of the β2-adaptin trunk, which collides sterically with Nef in the modeled conformation. It seems straightforward that this helix could pivot out of the way, but this has yet to be directly tested. The general proximity of Nef, and its N-terminal region in particular, to the membrane is similar to the model proposed for the AP-1:Nef complex that functions in MHC-I downregulation (*Jia et al., 2012*). However, the details of

**Table 2.** Functional importance of Nef-AP-2 interacting residues

| Nef residue | References | Interacting residues | Interactions | References |
|---|---|---|---|---|
| Q104 | This study | α K298*, K299 | Hydrogen bond | (Chaudhuri et al., 2009) |
| D108* | This study | α K298*, K299 | Salt bridge | (Chaudhuri et al., 2009) |
| P129 | This study | α R341* | Van der Waals | (Chaudhuri et al., 2009) |
| R134* | This study | Nef D175* | Nef core-to-loop internal salt bridge | This study; (Mattera et al., 2011) |
| E154 | (Lindwasser et al., 2008) | σ2 R61, R10 | Salt bridge | This study |
| N157 | (Lindwasser et al., 2008) | σ2 A63 | Van der Waals | (Mattera et al., 2011) |
| E160 | (Lindwasser et al., 2008) | σ2 R15, α R21 | Salt bridge | (Mattera et al., 2011) |
| N161 | (Lindwasser et al., 2008) | σ2 C99, L101 | Van der Waals | This study; (Mattera et al., 2011) |
| S163 | (Lindwasser et al., 2008) | σ2 N97 | Weak hydrogen bonds | |
| L164* | (Janvier et al., 2003; Lindwasser et al., 2008) | σ2 Y62, A63, L65, F67, V88, L91, V98, L103 | Hydrophobic | This study; (Mattera et al., 2011) |
| L165* | (Janvier et al., 2003; Lindwasser et al., 2008) | σ2 Y62, H85, V88, E89, N92 | Hydrophobic | This study; (Mattera et al., 2011) |
| S169 | (Lindwasser et al., 2008) | σ2 A63 | Van der Waals | (Mattera et al., 2011) |
| L170* | (Lindwasser et al., 2008; Jin et al., 2013) | σ2 Y62 | Hydrophobic | This study |
| H171 | This study; (Lindwasser et al., 2008) | σ2 A63 | Hydrogen bond to main chain carbonyl | This study; (Mattera et al., 2011) |
| M173 | (Lindwasser et al., 2008) | α Q301, V300; σ2 R60, Y62, H85 | Hydrophobic and nitrogen-sulfur hydrogen bond | This study |
| D174* | (Lindwasser et al., 2008) | Nef Q104 | Nef core-to-loop internal hydrogen bond to main chain amide | This study |
| D175* | (Lindwasser et al., 2008) | Nef R134* | Nef core-to-loop internal salt bridge | This study; (Chaudhuri et al., 2009) |
| P176 | (Lindwasser et al., 2008) | α R341*, E342 | Van der Waals | This study; (Chaudhuri et al., 2009) |
| E177* | This study; (Lindwasser et al., 2008) | α R341* | Salt bridge | This study; (Chaudhuri et al., 2009) |
| R178 | (Lindwasser et al., 2008) | α E342, σ2 N48 | Strong hydrogen bond, weak salt bridge | This study |

**Bold**: mutation inhibits binding. *Italics*: mutation has no effect on binding. Plain text: not tested. Only residues tested as single amino acid substitutions are included.

*Mutation inhibits CD4 downregulation activity. All residues tested as single amino acid substitutions except for K298/R341, which were tested together.

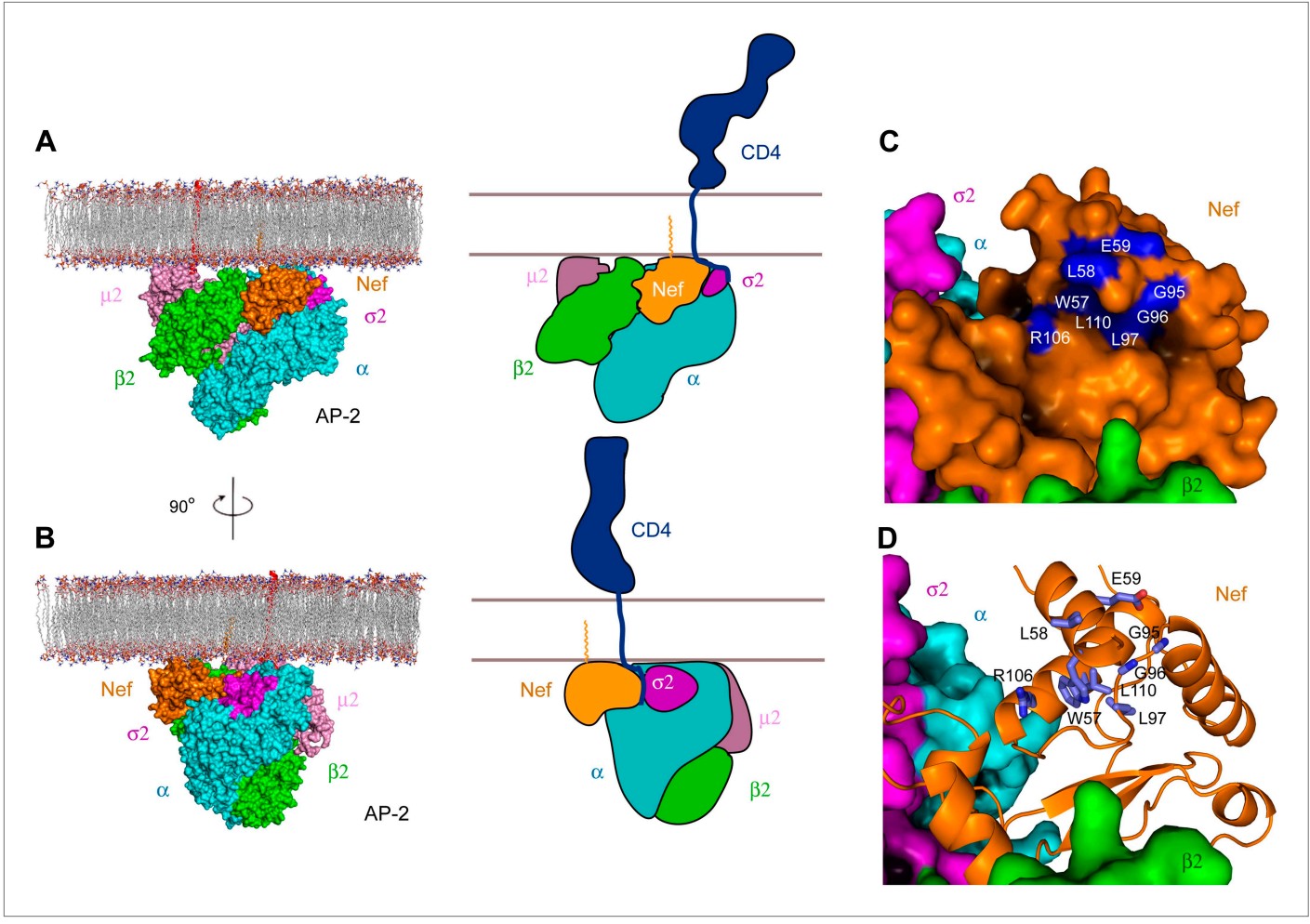

**Figure 7**. Docking of the unlocked AP-2:Nef complex to the membrane. (**A**) The unlocked conformation of AP-2 core bound to myristoylated Nef (orange). The AP-2 α–σ2:Nef structure was first aligned with the open conformation of AP-2 core structure (2XA7), and then docked on the membrane. The second view (**B**) is shown by rotating the first by 90°. Schematics are shown to the right of (**A**) and (**B**). (**C**) The surface of the AP-2:Nef complex as viewed from the membrane. Nef residues that are mapped by CD4 binding (*Grzesiek et al., 1996b*) are colored in blue. (**D**) Stick representation of Nef residues (blue) that interact with the CD4 cytosolic tail.

the molecular contacts between Nef and the membrane surface differ (*Figure 8*). This suggests that Nef has more than one way to interact with membrane surfaces.

The structural model, taken together with previous mapping of the CD4 binding site on Nef (*Grzesiek et al., 1996b*), suggests how CD4 binds to the AP-2 Nef complex on membranes. Residues of Nef helices H1 and H3 that are implicated in CD4 binding form one edge of a cavern 10 Å high and 30 × 30 Å across, with the membrane serving as the floor. The Nef binding site on CD4 comprises an approximate 17-residue tract (*Preusser et al., 2001*) that begins ~10 residues C-terminal to the end of the transmembrane helix. The entirety of the cavern is within 10 Å of the membrane surface. Thus localization of the CD4 tail within the cavern would be completely consistent with the proximity of the Nef binding site and transmembrane domain of CD4. The N-terminal loop of Nef is required for high-affinity binding to the CD4 cytosolic tail (*Preusser et al., 2001*). We therefore propose that the N-terminal loop of Nef folds around the CD4 tail within or at the edge of the cavern. The dissociation constants for CD4:Nef and AP-2:Nef are known separately, and both are approximately 1 μM. The affinity and kinetics of CD4 binding to the AP-2:Nef complex are unknown, but this model suggests that the binding would likely be tighter than for these isolated components, and the off rate correspondingly slower. This would contribute to delivery of CD4 to the ESCRT machinery and its ultimate degradation in the lysosome (*daSilva et al., 2009*).

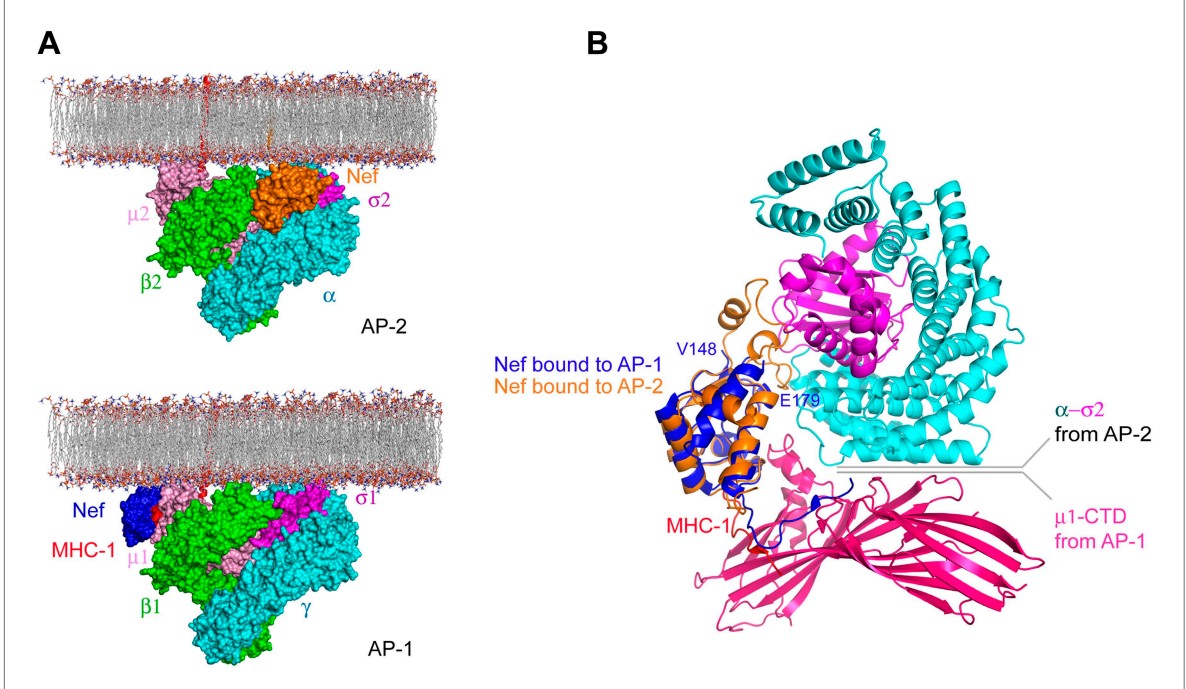

**Figure 8**. Nef uses different surfaces to bind to different regions of AP-1 and AP-2. (**A**) Nef binds to different subunits of AP-2 (top) and AP-1 (bottom), but docks onto the membrane in both cases. The MHC-I cytoplasmic domain (CD):Nef complex binds to the µ1 C-terminal domain (CTD) of the AP-1 core. One copy of AP-1 µ1-CTD:MHC-I-CD:Nef complex (*Jia et al., 2012*) (pdb: 4EN2) was aligned with the open conformation of AP-1 core structure (*Ren et al., 2013*) (pdb: 4HMY), and then the AP-1 complex was docked onto the membrane in the same orientation as shown for AP-2 in *Figure 7* and in the top panel. (**B**) Structural superposition of Nef (blue) as bound to the µ1 subunit of AP-1 upon Nef (orange) bound to the α–σ2 hemicomplex of AP-2 in this study.

The search for ways to combat the emergence of resistance to antiretrovirals, and to ultimately eradicate HIV, has led to an intense interest in targeting the interactions of HIV and host proteins (*Jaeger et al., 2011*). Host proteins, unlike viral proteins, are not susceptible to rapid mutation nor are they under selective pressure to resist therapy. The ideal host protein target site would be essential for the viral replication cycle, yet dispensable for host function. Such a site would also be 'groovy', that is to say, highly invaginated and capable of binding small molecules. The interaction surface complementary to the C-terminal turn segment of the Nef central loop would appear to fit this criterion. Indeed, a deep pocket in AP-2 directly below the binding site for Nef Arg178 appears to be under-utilized by Nef (*Figure 9*). Future analysis of the ternary AP-2:Nef:CD4 tail complex may reveal additional promising sites.

## Materials and methods

### Plasmid construction

Protein expression plasmids were constructed by restriction cloning. Rat α-adaptin (1–396) was subcloned as an N-terminal GST fusion together with rat σ2-adaptin into the pST39 polycistronic vector (*Tan, 2001*). A TEV protease cleavage site was introduced between the GST tag and α-adaptin. HIV-1 Nef (54–203) was subcloned into pHis2 (*Sheffield et al., 1999*) and expressed as a fusion with an N-terminal $His_6$ tag and a TEV cleavage site. All plasmids were verified by DNA sequencing.

### Protein expression and purification

The AP-2 α–σ2 hemicomplex was expressed in Rosetta2 cells (Novagen) and induced with 0.3 mM IPTG at 20°C overnight. The cells were lysed by sonication in PBS buffer, pH 7.4, 10% glycerol, 5 mM β-mercaptoethanol (BME), 5 mM EDTA, and a protease inhibitor cocktail (Sigma, St. Louis, MO). The clarified supernatant was first purified on GST Sepharose 4B resin (GE healthcare). After $His_6$-TEV cleavage at 4°C overnight, the sample was diluted in SP buffer A: 30 mM HEPES pH 7.4, 3 mM BME and then loaded onto a HiTrap SP HP 5 ml column (GE healthcare, Piscataway, NJ). Elution from the SP column

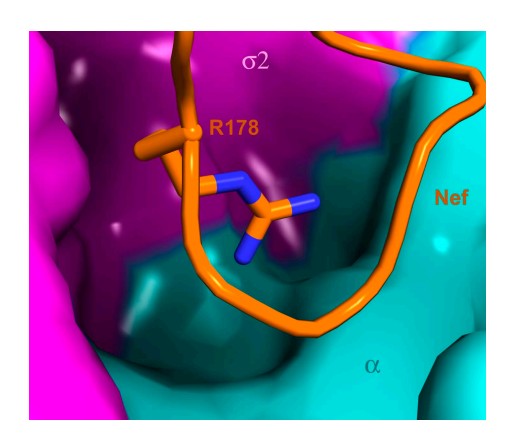

**Figure 9**. A highly concave pocket specific for the Nef interaction. Nef is shown in a stick model and the highly concave AP-2 surface is shown in the vicinity of Nef Arg178. This region has no known interactions with physiological cargoes.

was performed with a 70 ml linear gradient from 0–500 mM NaCl in SP buffer A. After each fraction was analyzed by SDS gel, the fractions were pooled and passed through 1 ml of GST resin and a Ni-NTA column (Qiagen, Valencia, CA) to capture the GST and His$_6$-TEV. This sample was further purified on a HiLoad 16/60 Superdex 200 column (GE healthcare) in 20 mM Tris pH 7.4, 200 mM NaCl, and 0.3 mM TCEP.

HIV-1 Nef constructs were expressed in BL21 (DE3) star cells (Invitrogen, Carlsbad, CA), and induced at 25°C overnight. The cell pellet was lysed by sonication and the lysate was loaded onto a Ni-NTA column in 50 mM Tris pH 7.4, 300 mM NaCl, 20 mM imidazole, 3 mM BME, 10% glycerol and protease inhibitor cocktail. The protein was eluted with 0.1 M imidazole, followed by TEV cleavage at 4°C overnight. After passing through Ni-NTA column to capture the cleaved His$_6$ tag, the sample was loaded to HiLoad 16/60 Superdex 75 column (GE healthcare) in the sample buffer.

## Isothermal titration calorimetry

Both TEV cleaved α–σ2 hemicomplex and His$_6$-tagged Nef(54-203) were purified by size-exclusion chromatography in the same ITC buffer of 20 mM Tris pH 7.4, 200 mM NaCl, 0.3 mM TCEP. The sample cell contained 0.2 ml of 40 μM α–σ2 hemicomplex, and Nef (600 μM) was added in 18 injections of 2.1 μl each. Data from Nef injections into buffer blanks were subtracted from sample data before analysis. Measurements were repeated three times and carried out on an itc200 instrument (MicroCal, Northampton, MA). The data were processed using Origin software (MicroCal). The binding constant ($K_d$) was fitted using a one-site model.

## Crystallization and crystallographic analysis

The α–σ2 hemicomplex was mixed with Nef (54–203) at a molar ratio of 1:1.2 in 20 mM Tris pH 7.4, 200 mM NaCl, 0.3 mM TCEP. Crystallization was carried out by sitting-drop vapor diffusion using an automated liquid-handling system (Mosquito, TTP LabTech, UK) at 288 K in 96-well plates. The optimized reservoir solution contained a mixture of 49 μl of Wizard I #29 (100 mM CHES pH 9.5, 200 mM NaCl, 10% PEG 8000, Emerald Bio, Bedford, MA) and 21 μl of 70% glycerol, adjusted to 0.2 mM inositol hexakisphosphate. The ratio of protein/precipitant in the drop was set at 2:1. The final crystal was obtained in 2–4 days by micro-seeding at 5 mg/ml α–σ2 hemicomplex. The crystals were soaked in the cryprotectant paratone-N (Hampton research, Aliso Viejo, CA) and frozen in liquid N$_2$.

Native data were collected from a single frozen crystal using a MAR CCD detector at beamline 22-ID, Advanced Photon Source. All data were processed and scaled using HKL2000 (HKL research, Charlottesville, VA). The crystal diffracted to 2.9 Å resolution, and belonged to space group P2$_1$2$_1$2$_1$ with unit cell dimensions a = 109.56 Å, b = 168.03 Å, c = 200.20 Å, α = β = γ = 90°. A molecular replacement solution was found using partial structures derived from the locked AP-2 core (PDB: 2VGL) (*Collins et al., 2002*) and from Nef/Hck-SH3 (PDB: 3REA) (*Breuer et al., 2011*) as search models with Phaser (*McCoy et al., 2007*). Model building and refinement were carried out using Coot (*Emsley et al., 2010*) and Phenix (*Adams et al., 2010*; *Table 1*). Structural figures were generated with PyMol (*DeLano, 2002*).

## Yeast 3-hybrid analysis

Y3H analysis was performed as previously described (*Chaudhuri et al., 2007*, *2009*). NL4-3 Nef or mouse tyrosinase cytosolic tail DNAs were subcloned into the pBridge vector (Clontech, CA) along with rat σ1 or σ2. Rat α and δ subunit DNAs were subcloned into the pGADT7 vector (Clonetech, CA). All the point mutants used in this study were generated by site-directed mutagenesis, using the QuikChange II XL (Agilent technologies, Santa Clara, CA). The canonical dileucine-containing tyrosinase tail construct

was included as a positive control for the formation of a functional complex, and the σ1 subunit of AP-1 and the δ subunit of AP-3 were included as negative controls for self-activation. The mutations were verified by DNA sequencing. The *Saccharomyces cerevisiae* HF7c strain was cotransformed with the indicated pairs of pBridge and pGADT7 constructs, using EZ Yeast Transformation Kit (MP biomedicals, Solon, OH). Double transformants were selected and grown on plates lacking Leu, Trp, and Met (+HIS) for 3 days, then the colonies from each transformant were normalized and plated on + HIS plates and plates lacking Leu, Trp, Met, and HIS (−HIS) with/without 3-AT (3-amino-1,2,4-triazole) for 4 days.

### Fluorescence-activated cell sorting analysis

FACS analysis was performed as described before (*Chaudhuri et al., 2009*). Wild-type or mutant NL4-3 Nef was subcloned into the pIRES2-eGFP vector (Clontech, CA). HeLa cells were co-transfected with pCMV-human CD4 and pIRES2-eGFP Nef wild-type or each mutant for 24 hr. The cells were then collected and stained with APC-conjugated anti-CD4 antibody and PE-conjugated anti-Transferrin receptor (TfR) antibody. The fluorescence was measured on a FACScalibur flow cytometer and analyzed by using CellQuest software (Becton Dickinson, Franklin Lakes, NJ). Only GFP positive cells were counted, and the inactive D174A, D175A Nef mutant was used as a negative control.

## Acknowedgements

This work was supported by a National Institute of General Medical Sciences grant (P50GM082250) to A Frankel and the Intramural program of the Eunice Kennedy Shriver National Institute of Child Health and Human Development and the Intramural AIDS Targeted Antiviral Program of the National Institutes of Health. Use of the Advanced Photon Source was supported by the U. S. Department of Energy, Office of Science, Office of Basic Energy Sciences, under Contract No. W-31-109-Eng-38.

## Additional information

### Funding

| Funder | Grant reference number | Author |
| --- | --- | --- |
| National Institute of General Medical Sciences | P50GM082250 | James H Hurley |
| Eunice Kennedy Shriver National Institute of Child Health and Human Development | HD001607-22 | Juan S Bonifacino |
| The Intramural AIDS Targeted Antiviral Program of the National Institutes of Health | | Juan S Bonifacino |

The funders had no role in study design, data collection and interpretation, or the decision to submit the work for publication.

### Author contributions

XR, Conception and design, Acquisition of data, Analysis and interpretation of data, Drafting or revising the article, Contributed unpublished essential data or reagents; SYP, Acquisition of data, Analysis and interpretation of data; JSB, JHH, Conception and design, Analysis and interpretation of data, Drafting or revising the article

## Additional files

### Major dataset

The following dataset was generated:

| Author(s) | Year | Dataset title | Dataset ID and/or URL | Database, license, and accessibility information |
| --- | --- | --- | --- | --- |
| Hurley JH, Bonifacino JS, Ren X, Park SY | 2013 | Crystal structure of AP-2 alpha/simga2 complex bound to HIV-1 Nef | 4NEE; http://www.rcsb.org/pdb/search/structidSearch.do?structureId=4NEE | Publicly available at the RCSB Protein Data Bank (http://www.rcsb.org/). |

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
