## [Decision Letter]

Thank you for sending your work entitled “How HIV-1 Nef Hijacks the AP-2 Clathrin Adaptor to Downregulate CD4” for consideration at *eLife*. Your article has been favorably evaluated by a Senior editor and 3 peer reviewers, one of whom is a member of the Board of Reviewing Editors.

The Reviewing editor and the other reviewer discussed their comments before we reached this decision, and the Reviewing editor has assembled the following comments to help you prepare a revised submission.

The authors describe the structure and associated biochemistry of the complex between HIV Nef and the AP-2 “hemi-core” – i.e., the first 396 residues of α with σ tucked into its right-angle bend. Overall, this work is straightforward, of high technical quality, and represents an important advance.

The crystal structure reveals how an otherwise unstructured loop within Nef contacts both of the two AP-2 subunits and it nicely explains a large body of mutagenesis data, both published and performed within this study. Several aspects of the work are of particular interest and importance, including:

1) The structure provides the first ordered picture of the large internal Nef loop, presumably because the structure only becomes ordered when it binds AP-2.

2) As the authors note, CD4 downregulation is the “classic” function of Nef, and this study explains the molecular basis for half of that interaction, explaining how Nef recruits the AP-2 complex to initiate assembly of a clathrin coated pit. The Nef interaction with CD4 remains to be characterized, but the current structure is a very nice step forward and the Nef-AP-2 interaction interface is sufficiently complex that it could not have been understood without a crystal structure.

3) The supporting biochemistry, together with previous mutational analyses, makes a compelling case for the relevance and completeness of the structure.

4) Knowledge of adaptor complex structures and membrane binding interactions enables a sensible model for the entire Nef-AP-2 complex, revealing how an ancillary cargo adaptor like Nef can cooperate with a generalized heterotetrameric adaptor.

5) The authors also make a “groovy” suggestion about an inhibitor site on AP-2 that might, if occupied, selectively interfere with Nef association.

The work is generally of high technical quality and no changes are required for publication in *eLife*, but we suggest that the authors consider the following issues.

1) Between the structure description and the descriptions of the mutations, the text is overly dense. For example, the comprehensive list of residues that line the “pocket walls”, the guided tour along the Nef loop and similar passages may prevent the more general reader from enjoying the paper. Figure 4 doesn't really help much because it shows the Nef interaction footprint, but doesn't show the positions of any residues. Table 2 is a nice summary of the different mutational analyses, but it doesn't explain the structural role of any of the mutated residues. The authors should consider expanding/modifying Figure 4 and Table 2 to highlight the relevant mutation sites (Figure 4) and briefly explaining the interactions that they make (Table 2). That would make the information more accessible, and also allow the authors to remove detailed descriptions of interfaces and contacts from the text.

2) In place of some of the structure description, the authors should consider adding a more sophisticated comparative analysis (and perhaps even a comparison figure) of the two Nef-AP complexes that have now been characterized structurally, i.e., their structure and the ternary complex of Nef with the MHC-I cytoplasmic domain and the μ1 subunit of AP-1 (Jia et al. NSMB 2012 19, 701).

3) A schematic version of Figure 7, placed alongside the current versions, based on real molecular surfaces (not replacing them), would be helpful.

4) The interaction between R178 of Nef and E342 of α-adaptin is interesting in part because it is specific for Nef; the mutation E342K in alpha adaptin affects the binding to Nef but not to the cellular protein tyrosinase, and this interaction is emphasized in Figure 8. Consequently, the authors should consider whether analyzing the effects of mutating R178 of Nef in the yeast three hybrid and CD4 downregulation assays would make a more complete story.

5) The authors have reported previously that the presence of the CD4 cytoplasmic tail stimulates Nef binding to the hemi-complex. Although the current structure does not include CD4, can the authors speculate further on how binding to CD4 might have this affect?

---

## [Author Response]

*1) Between the structure description and the descriptions of the mutations, the text is overly dense. For example, the comprehensive list of residues that line the “pocket walls”, the guided tour along the Nef loop and similar passages may prevent the more general reader from enjoying the paper*.

We agree with this good suggestion and the two sections mentioned were sharply truncated.

Figure 4
*doesn't really help much because it shows the Nef interaction footprint, but doesn't show the positions of any residues.*
Table 2
*is a nice summary of the different mutational analyses, but it doesn't explain the structural role of any of the mutated residues. The authors should consider expanding/modifying*
Figure 4
*and*
Table 2
*to highlight the relevant mutation sites (*Figure 4*) and briefly explaining the interactions that they make (*Table 2*). That would make the information more accessible, and also allow the authors to remove detailed descriptions of interfaces and contacts from the text*.

Figure 4 and Table 2 have been extensively annotated following this excellent suggestion.

*2) In place of some of the structure description, the authors should consider adding a more sophisticated comparative analysis (and perhaps even a comparison figure) of the two Nef-AP complexes that have now been characterized structurally, i.e., their structure and the ternary complex of Nef with the MHC-I cytoplasmic domain and the μ1 subunit of AP-1 (Jia et al. NSMB 2012 19, 701)*.

We appreciate the chance to highlight this important comparison, and we have added a new Figure 8 incorporating this suggestion.

*3) A schematic version of*
Figure 7*, placed alongside the current versions, based on real molecular surfaces (not replacing them), would be helpful*.

The suggested schematic panels have been added to Figure 7 to the right of the overall structural panels A and B.

*4) The interaction between R178 of Nef and E342 of α-adaptin is interesting in part because it is specific for Nef; the mutation E342K in alpha adaptin affects the binding to Nef but not to the cellular protein tyrosinase, and this interaction is emphasized in*
Figure 8*. Consequently, the authors should consider whether analyzing the effects of mutating R178 of Nef in the yeast three hybrid and CD4 downregulation assays would make a more complete story*.

Mutation of Nef R178 to Ala was already shown to impair binding to alpha-sigma2 in the yeast three-hybrid assay (Lindwasser et al., J Virol. 2008 February; 82(3): 1166–1174, cited in the paper), as indicated in Table 2. We haven’t tested whether this mutation inhibits CD4 down regulation; it could be done, but this would significantly delay the publication of the paper.

*5) The authors have reported previously that the presence of the CD4 cytoplasmic tail stimulates Nef binding to the hemi-complex. Although the current structure does not include CD4, can the authors speculate further on how binding to CD4 might have this affect*?

A new sentence addressing this has been added.